# LLM Prompting for Localization: English or Native Language in Multilingual Text Understanding

## Abstract

Large Language Models (LLMs) are widely used in many text understanding and related applications. However, it has been documented that not all languages are equally represented in the training data of these models. This raises important questions about the models' performance on given tasks when inputs and prompts are provided in various languages, as well as how prompt localization should be approached to maximize task performance across languages. In this study, we focused on text data of meeting transcripts, which often contain disfluencies, transcription errors, and non-standard language use, making understanding more challenging, especially in multilingual settings. To investigate whether English prompts could effectively achieve different tasks on conversational transcripts across languages without loss of performance, we translated and open-sourced 200 real-world meeting transcripts from the Topic-Conversation Relevance (TCR) dataset (Fan et al., 2024) into 15 non-English languages. We conducted three types of transcript-related tasks and compared the outputs of English and native language prompts using both LLM-based automatic evaluation and human judgment. This provided a robust assessment of the output quality in multilingual scenarios. Our results showed that English prompts could achieve performance comparable to native-language prompts for most languages and tasks. This suggests that instructions in English may be sufficient for scalable LLM-based localization of conversational transcripts, reducing the need for extensive language-specific adaptation.

## 1 Introduction

LLarge Language Models (LLMs) have become central to text understanding applications across a wide range of languages, powering features such as information extraction, summarization, and comprehension in real-world products. While LLMs have demonstrated impressive capabilities on general text, their effectiveness on conversational data — such as meeting transcripts — remains less explored, especially in multilingual and multicultural contexts. As LLMs are increasingly deployed in global products that process spoken interactions and collaborative discussions, understanding their behavior and performance on conversational text across different languages is crucial for effective localization and user experience.

In this study, we focused on three core text-related tasks that are fundamental to many LLM-based applications involving conversation data: text extraction, text summarization, and general text understanding. These tasks are particularly relevant in the context of meeting transcripts, where accurate extraction of key information, concise summarization, and robust comprehension are essential for downstream applications such as knowledge management, search, and analytics. Our investigation spanned 200 real-world meeting transcripts covering 15 diverse languages, including both high- and low-resource settings, providing a comprehensive view of LLM performance on authentic conversational content.

The motivation for our work stems from the need to empirically assess whether English prompts are sufficient for high-quality LLM performance in multilingual conversational settings. We would like to answer whether native-language prompts are necessary for optimal results in the localization pro-

cess. To address this, we systematically compared the effectiveness of English and native-language prompts across the three tasks, employing both LLM-based automatic evaluation and human judgment. Such an evaluation approach allowed us to benchmark LLM performance. It also enabled us to validate the reliability of automated metrics in multilingual contexts.

Our study made several key contributions to the field of LLM product localization and multilingual text understanding for conversational data. First, we expanded the TCR dataset by releasing all English transcripts in compact format and 200 selected meetings translated into 15 other languages. Second, we presented the first comprehensive evaluation of prompt language strategies for transcript-related tasks. We studied the areas of text extraction, summarization and understanding with meeting transcripts. Third, by combining LLM and human evaluation, we provided robust evidence that English prompts could achieve performance comparable to native-language prompts across most languages and tasks.

## 2 RELATED WORK

LLM localization has rapidly advanced the field of natural language processing, enabling a wide range of applications across languages. However, most LLM development and evaluation have focused on English, raising concerns about their effectiveness and fairness in multilingual and multicultural contexts (Myung et al., 2024; Lai et al., 2024). As LLMs are increasingly deployed globally, understanding their limitations and strengths in non-English settings has become a critical research area.

A substantial body of research has established that the language used in prompts is a key factor in LLM performance across languages. Early work by Lin et al. (2022) systematically studied prompt language effects using a large multilingual GPT-style model (XGLM). They found that English prompts often yielded higher zero-shot and few-shot accuracy than prompts in the target language, even for high-resource languages like Chinese and Hindi. This was attributed to the dominance of English in pre-training data and vocabulary, making English a strong "universal" prompt language. Later studies have confirmed this pattern. For example, Lai et al. (2023) showed that English prompts tend to outperform native-language prompts on a range of NLP tasks. Similarly, Ozsoy (2024) observed this trend in LLM-based recommender systems for English, Spanish, and Turkish.

To understand these effects, researchers have evaluated prompt language strategies across a diverse set of tasks. Lai et al. (2023) systematically compared English and native-language prompts on seven tasks, including part-of-speech tagging, named entity recognition, relation extraction, natural language inference, question answering, commonsense reasoning, and summarization. Ozsoy (2024) focused on recommendation tasks, while Mondshine et al. (2024) provided a systematic analysis of prompt translation strategies for extractive question answering, named entity recognition, summarization, and natural language inference. Wang et al. (2025) investigated cross-lingual prompting for structured data generation, specifically in text-to-SQL and text-to-MIP instance generation. Huang et al. (2023) evaluated cross-lingual-thought prompting on a broad range of tasks across 27 languages, and Myung et al. (2024) and Thellmann et al. (2024) conducted large-scale multilingual evaluations, confirming that prompt language effects are not uniform but vary by task, language, and domain.

Beyond simple prompt language comparisons, several works have explored more sophisticated strategies to improve multilingual LLM performance. Qi et al. (2022) proposed a prompt-based fine-tuning framework for cross-lingual natural language inference (NLI), reformulating XNLI as a cloze (fill-in-the-blank) task using cross-lingual prompt templates. By generating prompts in both English and the target language and enforcing prediction consistency, they achieved significant improvements for low-resource languages without needing extra parallel data. Similarly, Mondshine et al. (2024) showed that selectively translating only certain components of a prompt, such as the instruction or examples, can outperform both fully English and fully native-language prompts, especially in low-resource settings. Cross-lingual and mixed-language prompting approaches (Wang et al., 2025; Huang et al., 2023) have also been shown to enhance LLM reasoning and reduce disparities, suggesting that prompt design should be adapted to both the task and the language.

Recent studies have further advanced cross-lingual prompt design for in-context learning. Tanwar et al. (2023) found that using random English exemplars in prompts leads to misalignment in cross-

lingual in-context learning. They introduced X-InSTA, a prompt construction method that pairs each example in the source language with a semantically similar example translated into the target language. This alignment of contexts yielded substantial gains on multilingual text classification tasks across 44 language pairs, outperforming unguided prompting by a wide margin. Their results show that careful alignment of few-shot exemplars can mitigate the performance drop seen with naive multilingual prompting.

In addition to prompt strategies, several studies have highlighted the impact of language resource level, internal model bias, and cultural context on LLM performance. Benchmarks such as BLEnD (Myung et al., 2024) and Language Ranker (Li et al., 2025) consistently find that LLMs perform best in high-resource languages, with significant gaps persisting for low-resource languages. Internal analyses (Schut et al., 2024; Zhong et al., 2024) reveal that LLMs often process information in their dominant training language, even when prompted in another language, which can introduce bias and affect localization quality. Large-scale multilingual evaluations (Thellmann et al., 2024) confirm that both language family and resource level are key determinants of LLM effectiveness.

Advances in cross-lingual training and feedback (Lai et al., 2024) have shown that instruction tuning and human feedback in multiple languages can help democratize LLM capabilities and reduce performance gaps. However, the optimal approach may involve a combination of English and native-language elements, tailored to the specific task and language resource level.

Despite this progress, there remains a gap in the literature. Most studies have focused on structured or formal text, such as news, Wikipedia, or benchmark datasets, and have not addressed the unique challenges of conversational data like meeting transcripts. Few works have evaluated prompt strategies for extracting, summarizing, or understanding information from real-world, multi-party dialogue. This leaves open questions about how prompt language affects LLM performance on conversational tasks across many languages.

## 3 METHODS

In this study, we explored the possibility of developing English prompts on English data and applying the same prompts for localization. To create a fair comparison across languages, we translated the same set of English transcripts and prompts into 15 different languages. We also included a small set of meeting transcripts originally in other languages and translated them into English to create comparisons in the opposite direction. To determine whether our approach generalized across multiple transcript-related tasks, we conducted three types of experiments, covering text extraction, text summarization, and text understanding. The LLM used in this study was GPT-4.1.

### 3.1 TRANSCRIPT DATA

For our experiments, we selected 200 English meeting transcripts from the Topic-Conversation Relevance (TCR) dataset and translated them into 15 languages. In addition, we included 100 meeting transcripts that were originally collected in French and Chinese respectively, and translated them into English.

### 3.1.1 ENGLISH DATASETS

The TCR dataset contains more than 1,500 English meeting transcripts. These meetings are all real project and government meetings that capture real-life conversation styles and dynamics. We further expanded it by including 117 English meetings from the ELITR corpus (Nedoluzhko et al., 2022) with newly generated topic summaries by LLM.

Furthermore, because our study involved human evaluation and translation and both required complete sentences and sufficient context for a more accurate understanding, we created a compact version of every TCR transcript. This version concatenated consecutive utterances from the same speaker into a single block and updated the associated metadata (start time, end time, word count, etc.) accordingly. We also added a new field, `original_line_ids`, to each line so that every block could be mapped back to the corresponding lines in the original transcript.

For the experiments reported in this study, we selected 200 meetings from the TCR dataset and translated them into 15 languages. The set comprised 75 university team meetings from ICSI (Morgan et al., 2001), 117 project meetings from ELITR, and 8 government meetings from MeetingBank (Hu et al., 2023). These sources were chosen because they are representative of the most common types of meetings.

### 3.1.2 TRANSLATED DATASETS

We selected 15 target languages based on our application usage, representation, and the availability of localization support. The languages were represented by their IETF language tags in the final dataset and throughout this paper. The target languages were German (de), Spanish (es), Estonian (et), French (fr), Hebrew (he), Croatian (hr), Italian (it), Japanese (ja), Korean (ko), Latvian (lv), Portuguese (pt), Russian (ru), Albanian (sq), Turkish (tr), and Chinese-Simplified (zh-Hans).

To choose a publicly available translation service, we selected 6 meetings and translated them into 11 languages[1] with Azure AI Translator and DeepL respectively. We then presented the original English transcript and two anonymous translations from two different services side by side to native speakers for a quality comparison. The participants did not show systematic preferences for any of the services across languages; therefore, we chose Azure AI Translator for its ease of access. We then translated the 200 selected meetings from English into the 15 target languages using Azure AI Translator service.

In the translation procedure, we sent the compact versions of the English transcripts as lists of strings and retrieved the translated texts for the target languages. We kept the speaker names, start and end timestamps the same as the English version. We eliminated the word counts and cumulative word counts from the metadata, because the definition of the smallest linguistic unit differs across languages.

The expanded TCR dataset with the compact format and all translated files was open-sourced at `https://github.com/anonymous`. The translated files for different languages could be identified by the suffix of `_languageTag`.

### 3.1.3 AUTHENTIC NON-ENGLISH DATASET

To create a reverse comparison, in which the original meeting transcripts were non-English, we also included 100 French transcripts from the Clair dataset by Hunter et al. (2023) and 100 Simplified Chinese transcripts from the VCSum dataset by Wu et al. (2023).

### 3.2 EXPERIMENTS AND EVALUATIONS

We aimed to test the performance of English prompts and native-language prompts on different languages across different tasks. Specifically, we conducted three experiments that covered text extraction, text summarization, and text understanding.

For each task, the English prompt was developed on a subset of the English TCR dataset and additional internal data. To obtain high-quality versions of these prompts in the 15 target languages, we first translated them with an LLM and then had each translation reviewed and refined by the localization team. The team had detailed knowledge of the English prompts' intentions. They ensured that every translated prompt kept the exact structure of the corresponding English prompt. Each translation also accurately captured all information in the native languages.

### 3.2.1 E1. TEXT EXTRACTION - GOAL DETECTION

The first experiment was to extract text from the transcripts based on the prompt. We asked the LLM to extract meeting goals if the participants had discussed them in the transcript. Because the transcripts were often informal, with incomplete words and disfluencies, the LLM had the flexibility to rewrite the extracted goals in formal written language. If no goal was detected, the prompt asked the LLM to return a string of "no goal detected."

---

[1]The 11 languages selected based on availability of native speakers and services: German, Spanish, Estonian, French, Japanese, Korean, Latvian, Romanian, Turkish, Chinese-Simplified, Chinese-Traditional.

In the experiment, we first ran the English prompt on English transcripts (E/E) and used the outputs as the baseline. For each additional language, we applied the English prompt (E/N) and the native-language prompt (N/N) to the translated transcripts. The outputs from both the E/N and N/N runs were in the target language.

During evaluation, we first assessed the similarity of the outputs from E/N and N/N runs to those from E/E. To get a robust analysis, we used both human and LLM evaluations.

- Human evaluation: We conducted a crowd-sourcing task through Prolific. In the task, participants who were fluent in both the target language and English judged how similar an English item was against an item in the target language. The first item was always from the E/E experiment, and the other one was from either the E/N or the N/N run. Similarity was rated on a scale from 1 to 10. Outputs from 40 meetings in German, Spanish, French and Chinese were included in this crowd-sourcing task. Each comparison pair was rated by at least 8 qualified participants, who passed all trapping questions and gold question checks.

- LLM evaluation: We created a separate evaluation prompt that took one output from an E/E run and one from an E/N or N/N run and rated their similarity on the same 10-point scale used in the human evaluation. The evaluation prompt was applied to all 200 outputs of E/N and N/N runs for each language respectively.

For both human and LLM evaluations, we compared the similarity scores of "E/N versus E/E" and "N/N versus E/E" to determine whether the English prompt achieved performance comparable to that of the native-language prompts.

In addition, we applied the same LLM evaluations to compare the E/N and N/N outputs on the native French and Chinese dataset described in Section 3.1.3. With this test, we could better understand whether the performances differed on the authentic native dataset and the translated dataset.

### 3.2.2 E2. Text Summarization – Decisions and Open Issues

The second experiment focused on the task of text summarization. Based on the full meeting transcripts, the prompt asked the model to identify the decisions made and the open issues that remained unresolved. The outputs were two lists of strings: a list of decisions and a list of open issues. When there was no decision or open issue, the corresponding list was left blank.

We ran the experiments and evaluations using the same procedure described in Section 3.2.1. We applied E/E, E/N, and N/N runs on all 200 meetings for the summarization task across English and the 15 languages. As there were two output lists (decisions, open issues), there were 400 outputs in total for each language. We also conducted a crowd-sourced human evaluation on 72 randomly selected list pairs in German, Spanish, French and Chinese. The platform and procedure were the same as described in E1. For LLM evaluation, the same evaluation prompt as in Section 3.2.1 was applied to all 400 outputs of E/N and N/N runs for each language, respectively.

For this experiment, we also applied the prompts and LLM evaluations on the non-English dataset described in Section 3.1.3.

### 3.2.3 E3. Text Understanding - Topic-Conversation Relevance

The third experiment evaluated how relevant each agenda topic was to a given transcript. The prompt took a 15-minute transcript and a list of agenda topics as inputs. It returned a list of relevance scores of 0 to 10 for the topic list by analyzing how relevant each topic was to the transcript contents. This task represented the aspect of text understanding.

As in the previous two experiments, we first ran the E/E prompts, followed by the E/N and N/N prompts. Since each meeting transcript was divided into fixed 15-minute snippets, we ended up with 590 prompts per prompt language–input language pair.

Given that the outputs were lists of relevance scores, we did not require subjective evaluations of similarity for this experiment. Instead, we simply compared the mean absolute differences (MAD) between E/E and E/N, and between E/E and N/N. The smaller the MAD, the closer the performance was to the baseline E/E performance.

Since the authentic French and Chinese dataset did not have topic annotations, we did not run this experiment on them.

## 4 RESULTS

For each experiment, we focused on comparing the E/N and N/N performance. In experiments E1 (text extraction) and E2 (text summarization), we compared the similarity scores against the baseline results. The higher the similarity scores, the closer the performance was to the English-on-English baseline. In experiment E3 (text understanding), we compared the outputs' MAD against the baseline outputs. The smaller the MAD, the better the performance was.

### 4.1 E1. TEXT EXTRACTION RESULTS

In the text extraction experiment, the English prompt achieved a 0.176 higher LLM similarity score than the native prompt on the same translated native transcripts (Table 1). This is a 2.0% better similarity score. The delta is statistically significant. The score distributions are shown in Figure 1. The English prompt received more of the higher similarity scores (8 and 9) and far fewer low similarity scores in other buckets. The performance across different languages is shown in Table 6 in the Appendix. Among the 15 languages, 7 showed that English had statistically significantly better performance, and 1 showed worse performance. The remaining 7 languages showed that the English prompt and native prompt performed similarly on the tasks.

Table 1: Text extraction - LLM average similarity scores against English prompt / English inputs

| Prompt / Inputs Languages | Count | Avg. Score | std | Difference | p-value |
|---|---|---|---|---|---|
| English / Native | 3,000 | 8.540 | 1.954 | 0.176 | 0.000 |
| Native / Native | 3,000 | 8.364 | 2.357 | - | - |

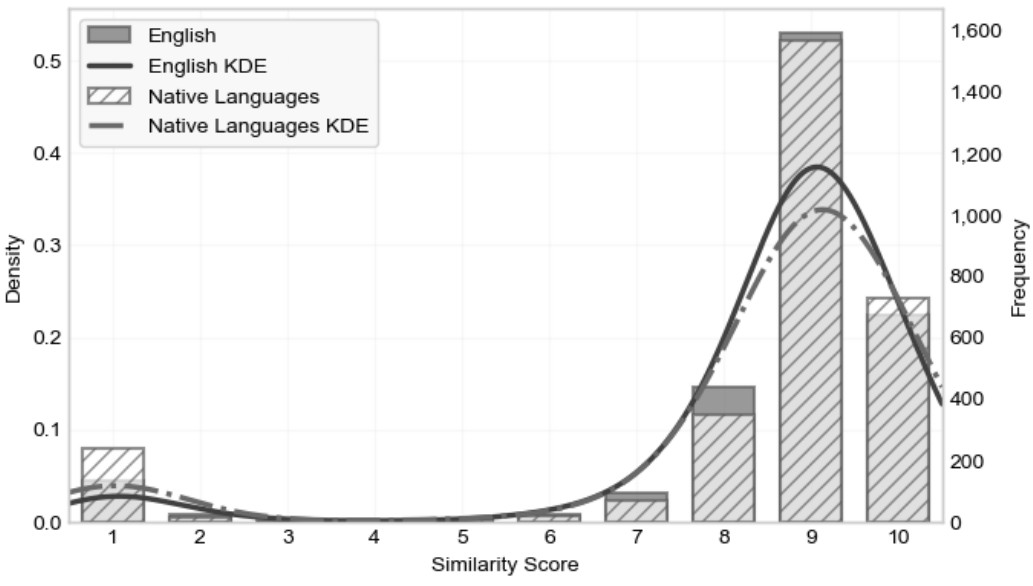

Figure 1: Text Extraction - Performance Comparison

The human evaluation results are listed in 2. None of the four languages showed any statistically significant difference between the English and native prompts on the native transcripts when comparing against the E/E ground truth.

We evaluated the similarity between the English prompt and the native prompt on the authentic French and Chinese datasets. The similarity scores were very high: 9.5 on the French dataset and

9.2 on the Chinese one. This supported the conclusion that the English prompt performed similarly as the native prompt even on authentic native datasets.

Table 2: Text extraction - human average similarity scores

| Language | English / Native | Native / Native | Difference | p-value | Stat. Sig. |
|---|---|---|---|---|---|
| French | 5.442 | 5.219 | 0.223 | 0.598 | - |
| German | 5.318 | 5.310 | 0.009 | 0.979 | - |
| Mandarin | 6.636 | 6.658 | -0.022 | 0.926 | - |
| Spanish | 5.442 | 5.219 | 0.223 | 0.598 | - |

## 4.2 E2. TEXT SUMMARIZATION RESULTS

However, for the text summarization experiment, the English prompt achieved a -0.185 lower LLM similarity score, which is a 2.6% relative drop compared with the native prompt. As shown in Table 3 and Figure 2, both the average similarity scores and the distribution are lower compared with those in E1. This suggests that the summarization task is generally harder than the extraction task. Also, a task involving more flexibility in terms of number of outputs, text rewriting, and grouping may have achieved lower scores due to the diversity in outputs. Furthermore, this experiment had higher variance across different languages. As shown in Table 7 in the Appendix, 8 out of the 15 languages showed that English had statistically worse performance; 2 languages showed that English had statistically better performance; and 5 languages showed that the differences were not statistically significant.

Table 3: Text summarization - LLM average similarity scores against English prompt / English inputs

| Prompt / Inputs Languages | Count | Avg. Score | std | Difference | p-value |
|---|---|---|---|---|---|
| English / Native | 6,000 | 6.894 | 1.640 | -0.185 | 0.000 |
| Native / Native | 6,000 | 7.078 | 1.578 | - | - |

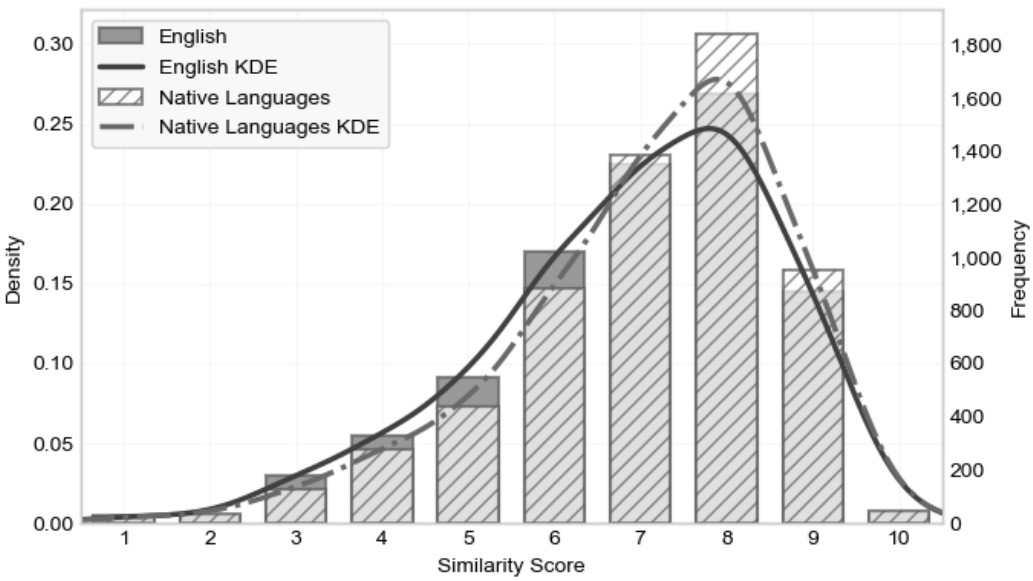

Figure 2: Text Summarization - Performance Comparison

For human evaluation, 3 out of the 4 languages did not show any statistically significant difference between the English and native prompts' performance. Spanish ratings suggested that the English prompt performed statistically better than the Spanish prompt by a +0.667 similarity score.

On the authentic French and Chinese datasets, the similarity scores for this task between the English and native prompts were 8.6 and 6.5, respectively. We investigated the low similarity on the Chinese dataset in more detail. The similarity score for open issues was relatively high (8.1), but the score for decisions was much lower (5.0), with 87% of the English prompts missing some decisions. Upon examining individual examples, we found that the English prompt performed as expected by including only in-meeting decisions, whereas the Chinese prompt also included conclusions from studies outside the meeting. Overall, the English and native prompts performed similarly in most parts of the summarization task. In cases where the two diverged, the English prompt performed as intended, while the native prompt may have needed more tuning.

Table 4: Text extraction - human average similarity scores

| Language | English / Native | Native / Native | Difference | p-value | Stat. Sig. |
|----------|------------------|-----------------|------------|---------|------------|
| French   | 5.266            | 5.229           | 0.037      | 0.873   | -          |
| German   | 4.480            | 4.180           | 0.300      | 0.280   | -          |
| Mandarin | 6.378            | 5.920           | 0.457      | 0.093   | -          |
| Spanish  | 6.226            | 5.559           | 0.667      | 0.010   | *          |

### 4.3 E3. TEXT UNDERSTANDING RESULTS

For the text understanding experiment (Table 5, Figure 3), the English prompt performed better than the native prompt by a margin of -0.069. Given that the output scores in this task were in the range of 0 to 10, the small difference between the two MADs suggests that the prompts actually achieved similar performance. Across all 15 languages, the advantage of English held across 13 languages, with 2 other languages not showing statistically significant differences.

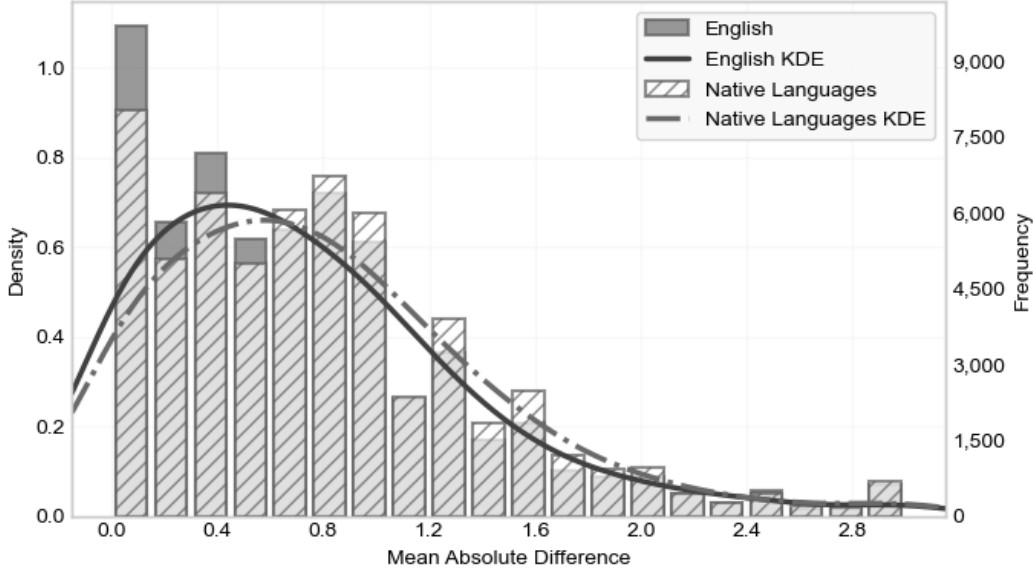

Figure 3: Text Understanding - Performance Comparison

Table 5: Text understanding - LLM mean absolute differences against English prompt / English inputs

| Prompt / Inputs Languages | Counts | Mean Abs. Diff | std | Difference | p-value |
|---|---|---|---|---|---|
| English / Native | 8,850 | 0.733 | 0.629 | -0.069 | 0.000 |
| Native / Native | 8,850 | 0.802 | 0.637 | - | - |

## 5 CONCLUSION AND NEXT STEPS

In this work, we systematically compared the effectiveness of English prompts and native-language prompts in 15 languages across three transcript-related tasks: text extraction, text summarization, and text understanding. Our results showed that a well–crafted English prompt generally performed on par with, and sometimes better than, prompts written in the target language. These findings indicated that a single English prompt could often be reused for multilingual localization, substantially reducing engineering effort and cost while maintaining quality. To encourage continued work, we expanded the TCR dataset by incorporating 3,000 translated meeting transcripts, created by translating 200 real meetings into 15 different languages.

In addition to these insights, several challenges remain. First, our multilingual inputs were created with machine translation. Higher-quality translations could help increase the overall performance against the baseline and potentially a more accurate comparison. Second, a meeting held in one language in its original culture setting may be very different from a meeting held in another culture even with the same meeting goal and agenda. The same conversation may not sound natural in another language even with perfect translation. Hence, making strict one-to-one comparisons is difficult. Third, most of our quality scores came from LLM-based evaluations, with only a small number of human evaluations collected. A larger number of human evaluations with text feedback could make the conclusion more robust and provide potential opportunities to improve.

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

# A APPENDIX

## A.1 LLM SIMILARITY SCORES BY LANGUAGE

Table 6: Text Extraction - Average Similarity Scores (LLM) against English Prompt / English Inputs by Language (Counts = 200 for each language)

| Language | Count | English / Native | Native / Native | Difference | p_value | Stat. Sig. |
|---|---|---|---|---|---|---|
| de | 200 | 8.510 | 8.360 | 0.150 | 0.204 | |
| es | 200 | 8.595 | 8.535 | 0.060 | 0.635 | |
| et | 200 | 8.590 | 8.220 | 0.370 | 0.030 | * |
| fr | 200 | 8.615 | 8.440 | 0.175 | 0.124 | |
| he | 200 | 8.525 | 8.535 | -0.010 | 0.942 | |
| hr | 200 | 8.495 | 7.960 | 0.535 | 0.007 | * |
| it | 200 | 8.645 | 8.320 | 0.325 | 0.001 | * |
| ja | 200 | 8.325 | 8.795 | -0.470 | 0.001 | * |
| ko | 200 | 8.550 | 8.580 | -0.030 | 0.772 | |
| lv | 200 | 8.710 | 8.270 | 0.440 | 0.000 | * |
| pt | 200 | 8.605 | 8.440 | 0.165 | 0.024 | * |
| ru | 200 | 8.590 | 8.340 | 0.250 | 0.016 | * |
| sq | 200 | 8.590 | 7.975 | 0.615 | 0.003 | * |
| tr | 200 | 8.380 | 8.280 | 0.100 | 0.561 | |
| zh-Hans | 200 | 8.370 | 8.405 | -0.035 | 0.719 | |

Table 7: Text Summarization - Average Similarity Scores (LLM) against English Prompt / English Inputs by Language

| Language | Count | English / Native | Native / Native | Difference | p_value | Stat. Sig. |
|---|---|---|---|---|---|---|
| de | 400 | 7.023 | 7.110 | -0.088 | 0.277 | |
| es | 400 | 7.145 | 7.383 | -0.238 | 0.004 | * |
| et | 400 | 6.970 | 7.198 | -0.228 | 0.008 | * |
| fr | 400 | 7.203 | 6.990 | 0.213 | 0.011 | * |
| he | 400 | 6.763 | 7.055 | -0.293 | 0.001 | * |
| hr | 400 | 6.913 | 6.955 | -0.043 | 0.625 | |
| it | 400 | 7.180 | 7.628 | -0.448 | 0.000 | * |
| ja | 400 | 6.713 | 7.453 | -0.740 | 0.000 | * |
| ko | 400 | 6.628 | 7.085 | -0.458 | 0.000 | * |
| lv | 400 | 6.758 | 6.518 | 0.240 | 0.006 | * |
| pt | 400 | 7.060 | 7.030 | 0.030 | 0.709 | |
| ru | 400 | 6.763 | 7.150 | -0.388 | 0.000 | * |
| sq | 400 | 6.470 | 6.540 | -0.070 | 0.447 | |
| tr | 400 | 6.673 | 6.950 | -0.278 | 0.001 | * |
| zh-Hans | 400 | 7.150 | 7.133 | 0.018 | 0.835 | |

Table 8: Text Understanding - Average Similarity Scores (LLM) against English Prompt / English Inputs by Language

| Language | Count | English / Native | Native / Native | Difference | p_value | Stat. Sig. |
|----------|-------|------------------|-----------------|------------|---------|------------|
| de | 590 | 0.719 | 0.779 | -0.060 | 0.001 | * |
| es | 590 | 0.664 | 0.784 | -0.120 | 0.000 | * |
| et | 590 | 0.732 | 0.781 | -0.048 | 0.004 | * |
| fr | 590 | 0.693 | 0.731 | -0.038 | 0.037 | * |
| he | 590 | 0.749 | 0.760 | -0.011 | 0.509 | |
| hr | 590 | 0.686 | 0.782 | -0.096 | 0.000 | * |
| it | 590 | 0.678 | 0.772 | -0.094 | 0.000 | * |
| ja | 590 | 0.737 | 0.834 | -0.097 | 0.000 | * |
| ko | 590 | 0.866 | 0.910 | -0.043 | 0.012 | * |
| lv | 590 | 0.750 | 0.827 | -0.077 | 0.000 | * |
| pt | 590 | 0.677 | 0.736 | -0.059 | 0.000 | * |
| ru | 590 | 0.770 | 0.834 | -0.065 | 0.002 | * |
| sq | 590 | 0.759 | 0.860 | -0.101 | 0.000 | * |
| tr | 590 | 0.732 | 0.823 | -0.091 | 0.000 | * |
| zh-Hans | 590 | 0.782 | 0.814 | -0.032 | 0.075 | |

## A.2 LLM USAGE

We used GPT-4.1 for all experiment tasks and automatic similarity evaluations as described in Section 3.2.

For paper writing, LLM was used only for grammar checking and language refinement. All substantive contents and analysis were performed by the authors.

