# OpenReview forum: "LLM Prompting for Localization: English or Native Language in Multilingual Text Understanding"
_ICLR.cc/2026/Conference — Submitted to ICLR 2026_

### Official Review · Reviewer_yvq4 · 2025-10-25

**Soundness:** 1
**Presentation:** 1
**Contribution:** 2
**Rating:** 2
**Confidence:** 4

**Summary:**

- The paper studies whether prompts for multilingual meeting transcript understanding tasks need to be localized into each target language, or whether an English prompt is “good enough,” even when the transcript itself is in another language.
- To study this, the authors take 200 English meeting transcripts from the TCR dataset and translate them into 15 target languages and also native French and Chinese transcripts to evaluate three downstream tasks: text extraction or goal detection, summarization, and topic-conversation relevance scoring.
- Key findings of the paper include for goal extraction, English prompts on non-English transcripts slightly better than native-language prompts. For summarization, native-language prompts slightly outperform English prompts overall but this is language-dependent. For topic relevance scoring, English prompts are marginally higher in agreement with English baseline than native prompts. Overall, this shows that a single well-designed English prompt can often be reused directly for multilingual transcript understanding, which could reduce localization cost in production systems.

**Strengths:**

- In terms of originality, he paper attempts to define “prompt localization” itself, in terms of “should we translate the prompt?” as the optimization target. This is practically important but also underexplored question in the LLM localization literature or deployment literature.
- The pipeline is at least partially rigorous: first, they evaluate both sides of localization: English → other languages and also other languages → English using native French and Chinese meetings. Also, the authors include both automatic evaluation (LLM as judge and MAD) and human evaluations.
- The experimental setup (E/E vs E/N vs N/N) is clearly defined and reused consistently across tasks. This repeated structure makes the paper easy to follow.
- The authors release a multilingual meeting transcript resource (200-way translations into 15 languages), which will likely be useful for future work on multilingual meeting assistants and evaluation of ASR/post-processing.

**Weaknesses:**

- The writing of the paper could be refined better, for example, currently there is no citations provided in the Introduction section.
- All experiments use a single proprietary model (GPT-4.1) both as the system under test and, for E1 and E2, also as the evaluator for similarity scoring. This might bias the alignment (e.g., GPT-4.1 may naturally like English prompts because it was trained on more English task instructions) and create shared model artifact, where using the same (or similar-family) model for both generation and judging can lead to reward hacking. The authors seem to be partially mitigating this problem with human eval, but human eval is limited to 4 languages and a fairly small subset of samples; most claims still lean on LLM-as-judge scores.
- We don’t see results from weaker or less Anglocentric models, or open models commonly deployed (Llama variants, Qwen, Mistral). If English prompting is only “good enough” when you’re using a frontier model with very strong multilingual capacity, that should be made explicit, otherwise the readers may incorrectly generalize the claim.
- If English prompting is only “good enough” when you’re using a frontier model with very strong multilingual capacity, that should be made explicit. Otherwise readers may incorrectly generalize the claim.
- The bulk of the non-English transcripts are machine-translated English meetings, not genuine meetings held in those languages or human translated. Since these datasets are naturally more diverse and noisy, it is difficult to 100% rely on machine translation outputs. Thus, if the underlying discourse structure, idioms etc are originally English, then of course an English-designed prompt might work on them. So the paper might be rather testing “Can GPT-4.1 read a machine translation of English speech using an English prompt?”, which is not necessarily the same as: “Can GPT-4.1 read a native Japanese or Spanish speech using an English prompt?” So the main headline that “you can just reuse English prompts for localization” may be overstated because most of the multilingual data are not authentically multilingual meetings.
- The paper frames better/worse performance in terms of how similar the multilingual output is to the English-on-English baseline output. That assumes that the English baseline output is ground truth, or at least high-quality. But there is no human annotation of actual task correctness independent of the English baseline. For example, we are not certain whether we actually extract all agenda goals that humans would say are goals? Or did we summarize the actual decisions that happened in the meeting, in the right language and culturally appropriate framing?

**Questions:**

- Is there any correlations between the machine translation quality of the transcripts and its performance in downstream tasks? (e.g., poor machine translation quality have lower performance)
- Did you try any open-weight multilingual models (e.g., Llama or Qwen) to see if the conclusion still holds? If not, can you at least discuss expected differences?
- For evaluation, did you try using a different judge model (not GPT-4.1) to reduce same-model bias?
- In reality, if companies adopt the “just use English prompts everywhere” strategy suggested by the paper, does that encode an English-centric definition of “goal,” “decision,” “topic relevance,” etc., that might miss any culturally salient commitments? Do you see any evidence of this phenomenon in your human evals? Maybe especially where Chinese annotators disagreed that this is already happening? If so, how could practitioners mitigate this issue?
- [Minor] Typo for the first word in the main paper: LLarge to Large.

---

### Official Review · Reviewer_hNLj · 2025-11-01

**Soundness:** 2
**Presentation:** 3
**Contribution:** 2
**Rating:** 4
**Confidence:** 4

**Summary:**

This paper studies whether English prompts are sufficient for multilingual conversational text understanding compared with native-language prompts. The authors translate 200 English TCR meetings into 15 languages (plus include authentic French/Chinese corpora for a reverse comparison), and evaluate three tasks: E1: text extraction (goal detection), E2: summarization (decisions & open issues), and E3: topic–conversation relevance scoring. Evaluations combine LLM-based similarity (vs. an English-on-English baseline) and limited human judgments (bilingual crowd workers on 4 languages). The authors' main finding is that English prompts perform as well or better for all of the tasks being evaluated. The authors also release compact versions of TCR transcripts and the 15-language translations.

**Strengths:**

- Whether English prompts suffice for multilingual deployments is a real product localization concern for LLM applications on noisy conversational data. The paper targets meeting transcripts, a domain underexplored relative to formal text.
- The work creates compact TCR transcripts and translates 200 meetings into 15 languages using a consistent pipeline; the data are said to be open-sourced. This resource could be useful if released with quality metadata.
- The findings are clear and actionable - generally speaking, using the English prompt performs reasonably similar to or even slightly better than language-specific prompts. The language-specific decomposition is also provided in the paper.

**Weaknesses:**

- The biggest concern I have is the limited scope of evaluation. Model-wise, only GPT 4.1 is evaluated, and I would expect to have at least one of the latest reasoning models. Tasks are also tailored to meetings; findings may not transfer to other conversational domains.
- The multilingual inputs are MT-generated. While the authors pilot-compared MT providers with native speakers on 6 meetings × 11 languages, they report “no systematic preferences” and then default to Azure—this is a light justification for a core data-creation step. No automatic MT quality (e.g., COMET/chrF) or error analysis is reported, and the paper itself notes this as a limitation.
- Human judgments cover only 4 languages and small subsets; the majority of claims rest on GPT-based similarity, which can encode biases that favor English or the specific model used.

**Questions:**

1. Can you provide the exact prompts (English & localized) for all tasks and the evaluation prompt for LLM-judge in the appendix/supplement, to enable reproduction?
2. What motivates the choice to use GPT-4.1 as the main model to evaluate? How is the conclusion sensitive to other models or using other models as the judge?

---

### Official Review · Reviewer_ejNo · 2025-11-02

**Soundness:** 2
**Presentation:** 2
**Contribution:** 2
**Rating:** 2
**Confidence:** 3

**Summary:**

The paper investigates whether English prompts can perform as well as native-language prompts in multilingual text understanding tasks. The authors focus on 200 meeting transcripts from the TCR dataset, translated into 15 languages, and test three tasks, text extraction, summarization, and understanding, using GPT-4.1. They compare English-on-English (E/E), English-on-native (E/N), and native-on-native (N/N) setups through both human and LLM-based evaluations. Results show that English prompts often perform comparably to native-language ones, with only minor differences across languages and tasks. The authors conclude that well-crafted English prompts may suffice for multilingual localization, reducing the need for language-specific prompt tuning.

**Strengths:**

- **Relevance:** The topic of prompt localization is timely and relevant for multilingual LLM use.
- **Scope:** The experiments cover 15 languages, including low-resource ones, providing valuable comparative data.
- **Dataset contribution:** The translated TCR transcripts are open-sourced, supporting future research.
- **Clarity of setup:** The three tasks (extraction, summarization, understanding) are well defined and follow a consistent protocol.

**Weaknesses:**

- **Limited novelty:** The main finding, that English prompts perform comparably to native ones, is unsurprising given prior literature on multilingual prompting.
- **Overreliance on translation:** Most data are machine-translated rather than authentically multilingual, which weakens claims about real-world generalization.
- **Shallow evaluation:** Heavy dependence on LLM-based scoring with little human verification introduces bias and circularity.
- **Insufficient analysis:** The paper lacks deeper error analysis or exploration of why some languages/tasks differ.
- **Presentation issues:** The text is long and dense, with oversized figures and redundant explanations.

**Questions:**

1. Report translation quality metrics and clarify how translation artifacts may affect results.
2. Include stronger baselines or controls, such as mixed-language or code-switched prompting, to test robustness.
3. Provide human evaluations for more than four languages to validate automatic scores.
4. Analyze which linguistic or typological factors explain differences across languages.
5. Add a concise discussion section that integrates findings, acknowledges limitations, and reflects on implications for LLM localization.

---

### Meta-Review · Area_Chair_zqUQ · 2025-12-10

**Summary:**

- The dataset relies too heavily upon machine translation which may not perform all that well on highly spontaneous and conversational meeting transcripts. [yvq4, hNLj, ejNo]
- The paper relies upon a single LLM as both the system under test and as the evaluators, which may lead to biased results. The attempt to mitigate this bias via human evaluation is quite limited. [yvq4, hNLj, ejNo]
- We don’t see results from weaker or less Anglocentric models, or open models commonly deployed (Llama variants, Qwen, Mistral). If English prompting is only “good enough” when you’re using a frontier model with very strong multilingual capacity, that should be made explicit, otherwise the readers may incorrectly generalize the claim. [yvq4]
- The paper frames better/worse performance in terms of how similar the multilingual output is to the English-on-English baseline output. That assumes that the English baseline output is ground truth, or at least high-quality. But there is no human annotation of actual task correctness independent of the English baseline. For example, we are not certain whether we actually extract all agenda goals that humans would say are goals? Or did we summarize the actual decisions that happened in the meeting, in the right language and culturally appropriate framing? [yvq4]

**Reviewer Concerns:**

There was no rebuttal, so none of the concerns have been addressed.

**Reviewer Scores:**

Not applicable because there was no rebuttal.

---

### Decision · Program_Chairs · 2026-01-26

Reject